# A structural equation model to access the regional public brands of agricultural products: Case of Chinese Yingde black tea

Jian Zhang[1,2], Xiaojun Ke[3]*, Songyu Jiang[4]

1 School of Entrepreneurship Management, Sanming University, Sanming City, Fujian Province, China, 2 School of Economics and Management, Guangzhou Nanyang Polytechnic College, Guangzhou City, Guangdong Province, China, 3 Guangzhou Institute of Science and Technology, Guangzhou City, Guangdong Province, China, 4 Rattanakosin International College of Creative Entrepreneurship, Rajamangala University of Technology Rattanakosin, Nakhon Pathom, Thailand

* drxjke@gzist.edu.cn

**Data Availability Statement:** All relevant data are within the manuscript and its Supporting Information files.

**Funding:** Guangdong Provincial Department of Education 2022 Key Areas (Rural Revitalization)

## Abstract

The regional public brand of agricultural products plays an important role in the development of agricultural economy. How to effectively build regional public brands for agricultural products is an urgent practical problem that needs to be solved in various regions. This study uses Chinese Yingde black tea as a case, aims to understand how government support, resource endowment, industrial clusters, and regional culture affect the regional public brand of agricultural products based on regional brand theory. We collected 416 valid sample data from practitioners related to Yingde black tea through an online survey questionnaire, and used structural equation modeling to obtain government support ($\beta = 0.196$, $p<0.005$), resource allowance ($\beta = 0.144$, $p<0.05$), industrial cluster ($\beta = 0.231$, $p<0.005$) and regional culture ($\beta = 0.335$, $p<0.005$) are positively related to reputation of regional public brands for agricultural products. Industrial clusters and regional culture play the mediating roles between government support, resource empowerment, and the reputation of regional public brands for agricultural products. The results breaks through the single influencing factor perspective of existing research and comprehensively analyzes the interrelationships of different influencing factors in the formation process of regional public brands for agricultural products. It has certain practical implications for the construction of regional public brands for agricultural products in China.

## 1. Introduction

The regional public brand of agricultural products is a behavior manifestation of the same or similar collective within a certain range [1]. Regional public brands represent the collective reputation and influence, and high visibility regional public brands are key factors in forming the attractiveness, loyalty, and reputation of agricultural products [2]. The construction of regional public brands for agricultural products is an important factor in promoting the development of agricultural economy, an important means to help rural populations overcome

Special Project: Research on the Construction and Countermeasures of Regional Public Brands for Agricultural Products in Guangdong Province (2022ZDZX4121); Key Research Platform Project of Guangdong Provincial Department of Education in 2023: Rural E-commerce Industry Education Integration Innovation Platform (2023CJPT022); The 2022 Education Reform Project of the Teaching Guidance Committee for Business and Trade Majors in Higher Vocational Colleges in Guangdong Province: Research on Information Technology Promoting the Transformation of Classroom Teaching in Vocational Colleges - Based on the Teaching Practice of Guangzhou Nanyang Vocational College of Technology (SM2022117).

**Competing interests:** The author(s) declared no potential conflicts of interest with respect to the research, authorship, and/or publication of this article.

poverty, and an important guarantee for achieving rural revitalization [3]. The Chinese government attaches great importance to the construction of regional public brands for agricultural products and has formulated relevant support policies multiple times [4].

According to the 2022 China High Quality Agricultural Products List released by the Third Party Evaluation Structure Panda Guide, there are 336 regional public brands of agricultural products in China on the list. This indicates that the regional public brand of Chinese agricultural products has achieved significant development through joint efforts from multiple parties, but brand building still needs further improvement. In 2022, Xinhui Chenpi ranked first in China's annual sales with a total of 1.9 billion US dollars [5]. In the same year, Champagne wine in France was $6.5 billion, the Washington apples in the United States were $2.4 billion, and the kiwi fruit in New Zealand was $2.3 billion [6]. These world-class regional public agricultural product brands not only exceed Xinhui Chenpi in terms of sales, but also have high market influence in the international market, with French Champagne wines, Washington apples from the United States, and kiwi from New Zealand all being sold globally [7]. The principal marketplace for Xinhui Chenpi predominantly resides within the confines of China, with merely a fractional segment being exported to areas in Southeast Asia characterized by significant Chinese communities [7]. Therefore, there is a certain gap between the regional public brands of Chinese agricultural products and world-renowned regional public brands in terms of product sales volume and brand influence.

In the context of China's rural revitalization strategy, how to enhance the influence of regional public brands of agricultural products and cultivate China's world-class well-known brands is not only a practical requirement for the high-quality development of China's agricultural economy, but also a necessary path for China to achieve a strong brand. The construction process of regional public brands for agricultural products is a systematic project that involves multiple influencing factors. Clarifying the interrelationships between different factors in the construction of regional public brands for agricultural products has become an urgent issue to be addressed in current practical work [8]. Nevertheless, the existing literature reveals a notable deficiency in research pertaining to the formation mechanisms underlying regional public brands for agricultural products. Concurrently, there is an observable scarcity of empirical findings on regional public brands specific to Chinese agricultural commodities. In light of these gaps, this paper aims to scrutinize Chinese Yingde black tea as the focal research entity. It endeavors to elucidate the genesis mechanisms of regional public brands within the agricultural sector, examine the determinant factors and their interrelations impacting such brands, and proffer recommendations for the establishment and reinforcement of regional public brands for agricultural products grounded on these insights.

Yingde black tea was cultivated by the Guangdong Provincial Tea Research Institute in 1959 through the introduction of Yunnan large leaf tea trees. After years of promotion and cultivation by local governments, research institutions, and agricultural enterprises, Yingde black tea has gradually grown into a well-known regional public brand of black tea both domestically and internationally. In 2006, Yingde black tea became a national geographical indication protection product in China. In 2010, the Yingde black tea trademark was awarded the National Geographical Indication Certification Trademark, and in 2014, it was included in the Sino European Geographical Indication Mutual Recognition Protection List. According to the 2023 China Tea Regional Public Brand Value Evaluation Report by the China Agricultural Brand Research Center, the brand value of Guangdong Yingde black tea is 594 million US dollars. Yingde black tea has gone through a process of development from nothing to something, from weakness to strength, and has a typical process of regional public brand construction for agricultural products. Therefore, this article takes Chinese Yingde black tea as the research object,

and the research results obtained can reflect the general laws of regional public brand construction of Chinese agricultural products, which has reference value for practical activities.

After the introduction, the next section provides an in-depth review of the relevant research on regional brands and regional public brands for agricultural products and elaborates in detail on the relationship between variables related to regional public brands for agricultural products. Based on this, the theoretical framework of this article is proposed. The next section provides a detailed introduction to the research design of this article. Subsequently, the fourth part elaborates on the results of empirical analysis based on sample data in this article. The climax of the research is the fifth part, which discusses the empirical analysis results to reveal the theoretical and practical significance of the research results. Finally, the sixth part introduces the research conclusions of this article and proposes ideas for future research.

## 2.Literature review

### 2.1. Theoretical approach

In the mid to late 20th century, product branding led enterprises to overcome the difficulties of manufacturing development, which sparked an explosive discussion of brand assets in the academic community. Branding strategy can not only be applied to products, but also to regional development [9]. Early scholars believed from a customer perspective that regional branding is the process of combining the effectiveness, emotions, and strategies of a region with the public's mind to generate special associations [10]. From an economic perspective, the regional brand refers to the total goodwill of enterprises and their affiliated brands formed within a regional scope, which have a considerable scale, strong manufacturing and production capacity, high market share and influence, and have the attribute of public goods [2]. The factors that affect the development of regional brands include regional planning, geographical environment, regional culture, and marketing [11]. With the maturity of regional brand theory, it has gradually been applied to research on national brands, city brands, regional brands, industrial cluster brands, and regional public brands for agricultural products [11]. Therefore, regional brand theory is the theoretical foundation of this article.

Drawing upon the foundational principles of regional brand theory, there are some comprehensive investigations encompassing the conceptual framework, attribute characteristics, and determinants influencing the establishment and development of regional public brands for agricultural products.

(1) The existing literature mainly elaborates on the connotation of regional public brands for agricultural products from four perspectives: the creator of the brand [12–14], the inherent advantages of brand development [15–17], The manifestation of brand identity [18], and the influence of the brand [10,19,20]. This research believes that the regional public brand of agricultural products is based on the unique resources, planting techniques, cultural history, and geographical features of the region, and has formed a certain level of popularity through historical accumulation. It is recognized by consumers and has become a public brand name that all producers in the region can enjoy [1]. (2) The attribute characteristics of regional public brands for agricultural products mainly include four types: brand attributes, regional attributes, public attributes, and asset attributes [21–23].

Understanding the factors affecting regional public brand of agricultural products, Wen-e et al. (2021) used Gannan navel oranges as an example to study that resource endowment is the core driving force for the development of regional public brands in agricultural products, but the role of regional culture is not significant [11]. However, Wei and Xiaobin (2020) concluded through long-term observation and research on the construction process of Wuchang rice that regional public brands are the result of long-term accumulation of a long history and

culture within the region [24]. A deep regional culture plays an important role in the formation of regional public brands, providing many opportunities and possibilities for brand development, using Taigu jujube as an example [25]. Whether regional culture affects the construction of regional agricultural product brands is an important controversial point. Industrial clusters play a fundamental role in the development of regional public brands for agricultural products, but the role of the government is not significant [26]. Moreover, Government's influence on regional public brands of agricultural products is limited [27]. However, Government support is the driving force behind the development of regional public brands for agricultural products [4]. The construction of regional public brands for agricultural products cannot be completely achieved through market transaction mechanisms, and the government needs to promote it through the formulation of relevant policies [28]. It can be seen that there is also some controversy over whether the government has played a positive role in promoting the construction of regional public brands for agricultural products.

Some scholars have explored the relationship between individuals and regional public brands of agricultural products from a social psychology perspective. Multifaceted interactions have a significant positive impact on farmers' willingness to participate in the co-creation of regional public brands for agricultural products, with psychological contracts serving as a mediating factor between these interactions and farmers' co-creation intentions [29]. There is a close relationship between regional public brands of agricultural products and consumers' emotional attitudes, with both exerting mutual influence on each other [30]. The greater the psychological distance between consumers and agricultural products, the higher consumers rate those products from that region [31]. Jian, D. et al.(2016) explored farmers' willingness to protect the agricultural ecosystem from a social psychology perspective, based on the Theory of Planned Behavior. They found that farmers' behavior is significantly influenced by their willingness to protect the ecosystem, which in turn is significantly affected by their attitudes, subjective norms, and perceived behavioral control [32]. The psychological factors of farmers pertain to the micro-level individual psychological changes [33]; however, this study primarily explores the mechanisms of regional public brand development for agricultural products from a macro-level perspective. Therefore, this study does not incorporate social psychological factors into the research process.

In summary, previous research has mainly focused on the impact of a single factor on regional public branding of agricultural products, with little exploration of the interrelationships between different factors. There is a lack of research on the formation mechanism of regional public brands, which is not conducive to providing comprehensive guidance for the construction of regional public brands. Case analysis and qualitative exploration are the main approaches, lacking sufficient empirical research, finally, there are certain differences in viewpoints among different research results.

It is obvious there are different views on the factors affecting regional public brands of agricultural products, existing literature mainly categorizes the influencing factors of regional public brands of agricultural products into four categories: resource endowment [15,26,34], industrial clusters [35–37], local governments [19,38,39] Humanities, History, and Culture [40–42]. Brand reputation is a concentrated reflection of consumer attitudes towards regional public brands, and is also considered the most important achievement in the construction of regional public brands [29]. Moreover, the regional public brand of agricultural products, as a public asset within the region, mainly affects the sales of related enterprise products in the region through brand reputation [43]. Therefore, this study chooses brand reputation as the benchmark variable to measure the effectiveness of regional public brand construction for agricultural products. Based on this, this article takes government support, resource empowerment, industrial clusters, and regional culture as independent variables, and the reputation of

regional public brands for agricultural products as the dependent variable, to study the formation mechanism of regional public brands for agricultural products. This article will answer the following questions: What are the interrelationships between different influencing factors in the construction process of regional public brands for agricultural products? Is there an intermediary effect in the construction process of regional public brands for agricultural products? How to develop a construction strategy for regional public brands of agricultural products based on regional characteristics?

## 2.2. Research hypotheses

Government support is mainly reflected in policy formulation, resource integration, industrial subsidies, and market supervision during the process of building regional public brands for agricultural products. The regional public brand of agricultural products is a special type of public brand that does not have exclusivity [44]. Therefore, the construction of regional public brands for agricultural products cannot be completely achieved through market transaction mechanisms, and the government needs to promote it through the formulation of relevant policies [28]. The government has a significant impact on the construction of regional public brands in market regulation and policy formulation [45]. Government support can play the following roles in the construction process of regional public brands for agricultural products: coordinating resource allocation within the region, supervising the business activities of enterprises within the region, guiding the development direction of regional public brands, and promoting the coordination of various stakeholders of regional public brands [4]. The government can have a direct impact on the development direction, speed, and mode of regional public brands of agricultural products by formulating policies [46]. Positive government policies can promote the development of regional resource endowments, provide high-quality public resources and preferential policies for related agricultural enterprises, attract upstream and downstream enterprises in the agricultural industry chain, and promote the development of relevant industrial clusters [44]. The development of industrial clusters will further strengthen government support for this industry and introduce more favorable policies [47]. The government helps promote the construction of regional culture by organizing exhibitions, celebrations, and food culture festivals for characteristic agricultural products [48]. The prosperity of regional culture will also lead the government to increase investment and promotion in the construction of this culture [49]. Therefore, Positive government policies can also promote the promotion of regional culture, contribute to its prosperity, and ultimately enhance the reputation of regional public brands for agricultural products [50]. Based on the above analysis, this article proposes the following three hypotheses.

H1: Government support has a positive impact on industrial clusters.

H3: Government support has a positive impact on regional culture.

H5: Government support has a positive impact on the reputation of regional public brands for agricultural products.

Agricultural products have a strong dependence on the natural environment [51]. Resource empowerment is an important foundation for the formation of regional public brands for agricultural products [41]. The unique climate, water sources, and soil generated by the special regional environment contribute to the cultivation of distinctive and high-quality agricultural products, and become a natural barrier that similar agricultural products in other regions cannot imitate [52]. Resource empowerment can also directly affect the quality, yield, and cost of agricultural products [8]. Therefore, resource empowerment contributes to the construction

of industrial clusters with unique agricultural products as the core, and enables the industrial cluster to gain resource advantages that cannot be imitated [53].Resource empowerment endows agricultural products with unique advantages, which helps cultivate the unique characteristics of regional public brands for agricultural products, thereby forming a unique brand image in the minds of consumers [11]. The production, operation, and consumption activities of agricultural products that arise from unique natural environments contribute to the formation of a unique regional culture [54]. Based on the above analysis, this article proposes the following three hypotheses.

H2: Resource empowerment has a positive impact on industrial clusters.

H4: Resource empowerment has a positive impact on regional culture.

H6: Resource empowerment has a positive impact on the reputation of regional public brands for agricultural products.

An industrial cluster for agricultural products is a group of enterprises engaged in the production, operation, or sales of agricultural products [53]. Industrial clusters contribute to the scale and intensive management of agricultural products, promote division of labor and cooperation among enterprises within the industrial chain, and thus generate economies of scale and synergies [7]. The scale and synergy effects of industrial clusters can promote division of labor and cooperation among regional enterprises, overcome the dispersion and uncertainty risks of individual enterprises participating in market transactions, and improve the competitiveness of enterprises [55].Enterprises within an industrial cluster cooperate and compete with each other, which helps to promote technological innovation and management change within the cluster, thereby promoting the improvement of regional agricultural product management efficiency [50]. Industrial clusters can facilitate the interaction and influence of enterprises within a region, and their spillover effects make the overall value of enterprises in the region significantly greater than the accumulated value of individual enterprises, thereby promoting the rapid development of regional public brands [26]. Therefore, industrial clusters play an important role in promoting the construction of regional public brands for agricultural products and promoting the improvement of the reputation of regional public brands for agricultural products. Based on this, this article proposes the following assumptions.

H7: Industrial clusters have a positive impact on the reputation of regional public utility brands for agricultural products.

Based on assumptions H1 and H7, this article proposes the following assumptions:

H9: Industrial cluster plays a mediating role between government support and the reputation of regional public brands for agricultural products.

Based on assumptions H4 and H7, this article proposes the following assumptions:

H11: Industrial clusters mediate the relationship between resource empowerment and the reputation of regional public brands for agricultural products.

The process of long-term cultivation, processing, sales, and consumption of characteristic agricultural products will gradually give rise to unique production methods, manufacturing techniques, as well as consumer culture and folk customs, and gradually form regional culture with regional characteristics [56]. A deep regional culture plays an important role in the formation of regional public brands, providing many opportunities and possibilities for brand development [25]. The process of year-round production, sales, and consumption of characteristic agricultural products within the region will form a unique knowledge and culture, and

integrate into regional public brands, which helps consumers form cultural awareness convergence towards regional public brands [57]. Therefore, the unique regional culture endows the regional public brand of agricultural products with unique cultural connotations, which helps to build differentiated brand identification for regional public brands and improve the reputation of regional public brands of agricultural products [58]. Based on this, this article proposes the following assumptions.

H8: Regional culture has a positive impact on the reputation of regional public brands for agricultural products.

Based on assumptions H3 and H8, this article proposes the following assumptions:

H10: Regional culture mediates the relationship between government support and the reputation of regional public brands for agricultural products.

Based on assumptions H4 and H8, this article proposes the following assumptions:

H12: Regional culture mediates the relationship between resource empowerment and regional public brand reputation of agricultural products.

## 2.3. Model construction

Based on the above assumptions about the relationship between government support, industrial clusters, resource empowerment, regional culture, and the reputation of regional public brands for agricultural products, this article constructs an theoretical framework of regional public brands for agricultural products. As shown in Fig 1.

## 3.Research method

### 3.1. Data collection

The geographical protection range of Yingde black tea is from longitude 112° 45′ to 113° 55′ E and latitude 23° 50′ to 24° 33′ N. The protection scope includes four major tea production areas in Eastern Ying, Central Ying, Northwestern Ying, and Southwestern Ying of Yingde City, China. Therefore, this article takes the four major production areas of Yingde black tea as the investigation area. The survey targets government officials, black tea associations, black tea related enterprises, black tea research institutes, and black tea growers in the four major production areas. Based on the number of relevant personnel in the four major tea producing regions and the random distribution requirements of the survey samples, this article randomly selected 150, 100, 150, and 200 people to fill out the survey questionnaire in the four tea producing regions. Before conducting the survey, we screened 20 college students majoring in business and provided them with skill training through questionnaire surveys. We then had these 20 college students distribute a total of 600 survey questionnaires and 565 questionnaires were received back. After sorting, we obtained 416 valid questionnaires, with a valid questionnaire response rate of 69.3%.

### 3.2. Instrument

By organizing and analyzing existing research literature, a mature scale with high citation rates was adopted to design the survey questionnaire for this article. Firstly, 65 survey subjects were randomly selected within the geographical protection area of Yingde black tea, and a small sample test was conducted on the survey questionnaire. Based on the results of small sample testing, the survey questionnaire was optimized and some items that did not meet the requirements were removed. The final survey questionnaire consisted of 26 items. The survey

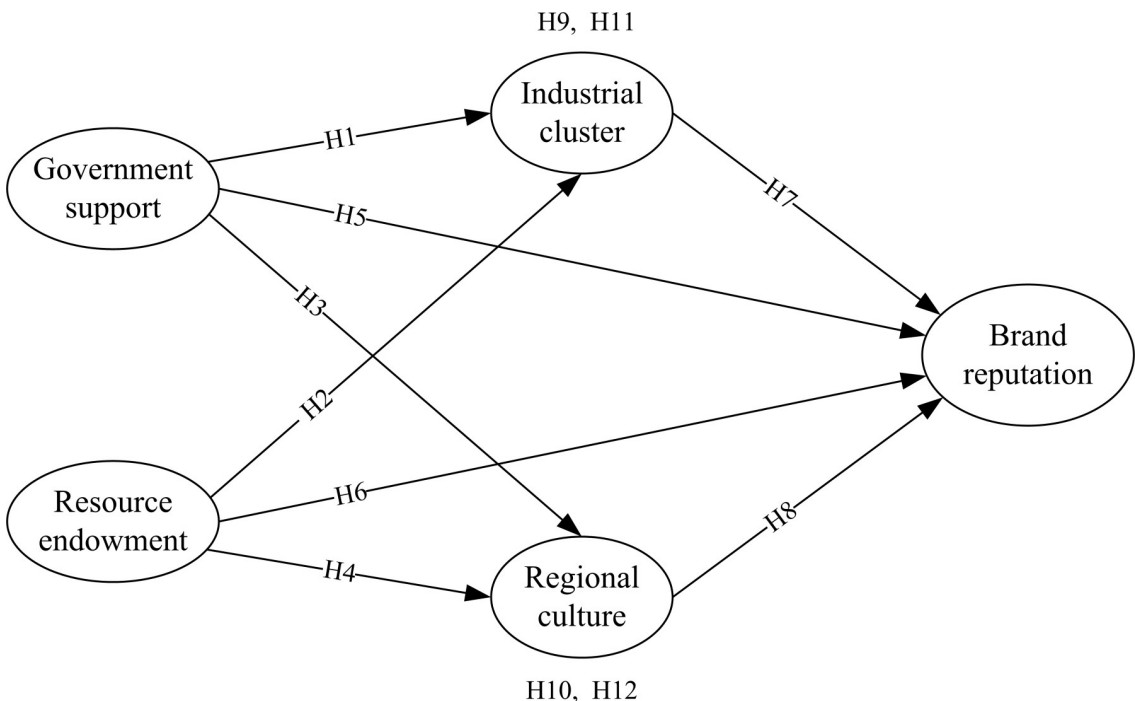

**Fig 1. Theoretical framework of regional public brands for agricultural products.**

questionnaire in this article consists of three parts: the first part is the basic information of the respondents, with a total of four questions, including their gender, occupation, working hours, and understanding of Yingde black tea [26]. The second part is a survey on the impact of government support, resource empowerment, industrial clusters, and regional culture on the reputation of regional public brands in agricultural products, with a total of 18 questions. The third part is a brand reputation survey that reflects the effectiveness of regional public brand construction for agricultural products, with a total of four questions. The survey questionnaire adopts the Likert seven scale measurement method, with 1 indicating strongly disagree, 2 indicating disagree, 3 indicating somewhat disagree, 4 indicating neutral, 5 indicating somewhat agree, 6 indicating agree, and 7 indicating strongly agree.

The measurement of latent variable government support refers to the questionnaires of Jun and Xin (2014), Jiexian and Shaofeng (2018), mainly measuring the government's impact on regional public brands of agricultural products in industrial policies, financial support, market supervision, strategic planning, and public services [59,60]. The measurement of resource empowerment refers to the questionnaires of Feilong et al. (2021), Chenglin and Xueping (2016), mainly measuring the impact of unique geographical conditions, black tea varieties, production area planning, and planting techniques on regional public brands of agricultural products [61,62]. The measurement of industrial cluster refers to the questionnaires of Yueli and Fang (2013) and Xueyi and Xinmao (2011), mainly measuring the impact of an industrial cluster's industrial chain, industrial structure, logistics cooperation, technical cooperation, and marketing cooperation on regional public brands of agricultural products [8,63]. The measurement of regional culture refers to the questionnaires of Shengzu et al. (2008), Weihong and Shengcheng (2021), mainly measuring the impact of regional food culture, planting culture, cultural activities, and cultural integration on regional public brands of agricultural products [64,65]. The measurement of the reputation of regional public brands for agricultural products

was based on the questionnaires of Hanyu and Huang (2016) and Jiali (2017). The construction effect of regional public brands for agricultural products was observed by measuring their popularity, reputation, customer loyalty, and brand image [66,67]. The specific measurement scales for each latent variable are shown in Table 1.

## 3.3. Data analysis

Structural equation modeling is a multivariate statistical method based on variable covariance matrix for factor analysis and path analysis. It is mainly used to explore the potential

**Table 1. Scale of the formation mechanism of regional public brands for agricultural products.**

| Variable | Observed variable |
|---|---|
| Respondent | What is your gender? (1)Male (2)Female |
|  | Do you know about the regional public brand of Guangdong Yingde black tea? Understanding (2) I don't understand (If you choose this option, the following questions do not need to be filled out again.) |
|  | What is your profession? (1) Government staff (2)Employees of Yingde black tea related enterprises (3) Yingde black tea growers (4) Researchers related to Yingde black tea (5) Staff of Yingde Black Tea Industry Association (6)Other |
|  | How long have you been engaged in your current job? (1)1-5 years (2)6-10 years (3)11-15 years (4)16-20 years (5)Over 20 years |
| Government support | **PS1:** The Yingde City Government has implemented policies aimed at fostering the growth of the Yingde black tea industry. |
|  | **PS2:** The Yingde City Government has allocated financial resources to bolster the advancement of the Yingde black tea industry. |
|  | **PS3:** The Yingde City Government has enhanced oversight and safeguarding measures for the black tea market. |
|  | **PS4:** The Yingde City Government has devised a strategic blueprint for advancing the black tea industry. |
|  | **PS5:** The Yingde City Government has intensified efforts in expanding public services and infrastructure development. |
| Resource endowment | **RE1:** The Yingde region boasts distinctive geographical features conducive to the cultivation of black tea. |
|  | **RE2:** The Yingde region is renowned for its distinctive variety of black tea. |
|  | **RE3:** The Yingde region has established scientifically planned black tea production zones guided by meteorological indices. |
|  | **RE4:** The Yingde region has pioneered scientific techniques for black tea cultivation. |
| Industrial cluster | **IC1:** The Yingde region boasts a comprehensive black tea industry ecosystem. |
|  | **IC2:** The Yingde region features a diversified structure within its black tea industry. |
|  | **IC3:** There is logistical and distribution collaboration among black tea-related enterprises in the Yingde region. |
|  | **IC4:** There is collaborative technological innovation among black tea-related enterprises in the Yingde region. |
|  | **IC5:** There is collaborative marketing communication and cooperation among black tea-related enterprises in the Yingde region. |
| Regional culture | **RC1:** The Yingde region boasts a distinctive culinary culture centered around black tea. |
|  | **RC2:** The Yingde region has a rich and enduring tradition of black tea cultivation. |
|  | **RC3:** The Yingde region frequently hosts distinctive cultural events celebrating black tea. |
|  | **RC4:** The Yingde region has seamlessly integrated black tea culture into its urban atmosphere. |
| Brand reputation | **BR1:** Yingde black tea enjoys widespread brand recognition. |
|  | **BR2:** Yingde black tea possesses a distinctive brand identity. |
|  | **BR3:** Yingde black tea receives favorable brand evaluations. |
|  | **BR4:** Yingde black tea boasts a large base of loyal customers who make repeat purchases. |

relationships between multiple variables and comprehensively analyze the interaction mechanisms between variables [68]. This article studies the formation mechanism of regional public brands for agricultural products, exploring the interaction between government support, resource endowment, industrial clusters, regional culture, and regional public brands for agricultural products. The SEM framework consists of two main components: the measurement model and the structural model [9]. Therefore, the data processing in this article adopts the method of structural equation modeling. Based on the collected 416 sample data, this article first conducted reliability and validity tests on the scale, and then used AMOS24 software to construct a structural equation model for regression analysis to test the research hypotheses.

The measurement model elucidates how latent constructs (represented by ξ) are measured by observed variables (X), incorporating factor loadings (λ) and measurement errors (δ). Mathematically, this is expressed as [69]:

$$X = \lambda \xi + \delta.$$

On the other hand, the structural model delineates the relationships between latent variables (ξ) and the ultimate dependent variable (Y) through structural coefficients (β) and structural errors (ε), denoted as [69]:

$$Y = \beta \xi + \varepsilon.$$

The parameters (λ, β, δ, ε) are estimated through statistical techniques, often employing maximum likelihood estimation [69]. Assessment of model fit involves various fit indices such as chi-square, comparative fit index (CFI), and root mean square error of approximation (RMSEA).

It is crucial to acknowledge that the specifics of SEM formulations may vary based on the nature of the model (e.g., confirmatory factor analysis) and the software employed for estimation [69].

## 4. Data analysis results

### 4.1. Describe statistical analysis

This article collected 416 valid samples, with males accounting for 64.183% and females accounting for 35.817%, which is consistent with the data of practitioners related to Yingde black tea [7]. The distribution proportion of respondents from government, black tea association, black tea research institute, black tea related enterprises, and black tea growers is relatively balanced, which meets the requirements of this survey and research. 86.538% of respondents have been working for more than 5 years and are familiar with the situation of Yingde black tea. The specific data of the sample is shown in Table 2.

### 4.2. Scale reliability test

This article is based on the collected data and uses Spss24 software to conduct reliability analysis on the survey scale. The Cronbach Alpha coefficient of the total scale is 0.919, which is greater than 0.7 [70]. The correlation coefficients between all observed variables after correction and the total are all greater than 0.4, and the Cronbach Alpha coefficients after item deletion are all greater than 0.9, which are all lower than the Cronbach Alpha coefficients in the total table [70]. Reliability analysis was conducted separately on each latent variable, and it was found that the Cronbach Alpha coefficients of all latent variables were greater than 0.8, and the Cronbach Alpha coefficients of each item after deletion were all greater than 0.7, and all were smaller than the corresponding Cronbach Alpha coefficients of the latent variables [70]. The

**Table 2. Describes the statistical analysis results.**

| Project | Category | Number | Percentage |
|---|---|---|---|
| Gender of the respondents | male | 267 | 64.183% |
| | female | 149 | 35.817% |
| The occupation of the respondent | government staff | 75 | 18.029% |
| | Employees of Yingde black tea related enterprises | 93 | 22.356% |
| | Yingde black tea growers | 80 | 19.231% |
| | Researchers related to Yingde black tea | 65 | 15.625% |
| | Staff of Yingde Black Tea Industry Association | 98 | 23.557% |
| | other | 5 | 1.202% |
| The working hours of the respondents | 1–5 years | 56 | 13.462% |
| | 6–10 years | 89 | 21.394% |
| | 11–15 years | 102 | 24.519% |
| | 16–20 years | 93 | 22.356% |
| | Over 20 years | 76 | 18.269% |

specific inspection results are shown in Table 3. Based on the above analysis, it can be concluded that the scale and observation indicators in this article have strong credibility.

## 4.3. Scale validity testing

This article conducted exploratory factor analysis on the survey scale using Spss24 software, and found that the KMO coefficient of the total scale was 0.918, greater than 0.7, and the Bartlett spherical test showed a significance of 0.000 [41]. Therefore, the sample data is suitable for conducting factor analysis. When using principal component analysis for factor analysis in this article, 5 factors were identified, which is consistent with the number of latent variables assumed in this article. After using Kaiser standardized orthogonal rotation, the percentage of squared variance of the rotational load is obtained. The higher the cumulative percentage, the higher the degree of variation of the data explained by the factor. Generally, if it exceeds 60%, it meets the requirements [53]. The total variance explained by factor analysis in this article is 67.146%, which meets the requirements. The common factor analysis of variance shows that the factor coefficients of all observed variables are greater than 0.5 [53]. Therefore, the explanatory power of the scale's factors is within an acceptable range.

This article also conducted confirmatory factor analysis using AMOS 24 software. The smaller the ratio of chi square degrees of freedom ($x^2/_df$), the higher the model's adaptability. Usually, a value less than 3 indicates good adaptability [11]. When the root mean square error of approximation (RMSEA) is less than 0.08, it indicates good adaptation [11]. The closer other indicators such as Comparative Fit Index (CFI) and Relative Fit Index (TLI) are to 1, the better the fit of the model. A value greater than 0.9 indicates better adaptation [11]. This article uses sample data to conduct confirmatory factor fitting analysis and finds that the ratio of chi square degrees of freedom is 1.786, which is less than 3; CFI, TLI, GFI, AGFI, and IFI are all greater than 0.9; RMSEA is 0.044 less than 0.08. Therefore, overall, the confirmatory factor model fits well.

This article uses factor loading, combined reliability (CR), and mean variance extraction (AVE) as the evaluation criteria for aggregate validity [7]. When the factor loadings of each latent variable are greater than 0.5, the CR value is greater than 0.7, and the AVE value is greater than 0.5, it is considered that the aggregated validity is good [7]. This article conducted an aggregate validity test using sample data and found that the factor loading of each observed

**Table 3. Reliability and validity test results.**

| Variable | Item | Clone Bach Alpha after deleting item | Revised item and total correlation | Factor Loading | Cronbach's Alpha | CR | AVE |
|---|---|---|---|---|---|---|---|
| Government support | PS1 | 0.822 | 0.713 | 0.776 | 0.860 | 0.863 | 0.561 |
| | PS2 | 0.856 | 0.575 | 0.647 | | | |
| | PS3 | 0.835 | 0.663 | 0.714 | | | |
| | PS4 | 0.839 | 0.645 | 0.696 | | | |
| | PS5 | 0.798 | 0.797 | 0.889 | | | |
| Resource endowment | RE1 | 0.846 | 0.642 | 0.720 | 0.859 | 0.861 | 0.608 |
| | RE2 | 0.814 | 0.722 | 0.784 | | | |
| | RE3 | 0.798 | 0.758 | 0.848 | | | |
| | RE4 | 0.824 | 0.698 | 0.762 | | | |
| Industrial cluster | IC1 | 0.819 | 0.566 | 0.671 | 0.834 | 0.837 | 0.508 |
| | IC2 | 0.815 | 0.585 | 0.637 | | | |
| | IC3 | 0.801 | 0.635 | 0.709 | | | |
| | IC4 | 0.792 | 0.666 | 0.728 | | | |
| | IC5 | 0.776 | 0.724 | 0.808 | | | |
| Regional culture | RC1 | 0.793 | 0.616 | 0.686 | 0.821 | 0.828 | 0.549 |
| | RC2 | 0.793 | 0.605 | 0.682 | | | |
| | RC3 | 0.776 | 0.645 | 0.744 | | | |
| | RC4 | 0.738 | 0.728 | 0.839 | | | |
| Brand reputation | BR1 | 0.829 | 0.738 | 0.817 | 0.871 | 0.872 | 0.630 |
| | BR2 | 0.842 | 0.709 | 0.774 | | | |
| | BR3 | 0.833 | 0.732 | 0.802 | | | |
| | BR4 | 0.835 | 0.723 | 0.782 | | | |

variable were all greater than 0.5; The AVE values extracted from the average variance of each latent variable are between 0.508 and 0.630, all of which pass the standard of greater than 0.5; The combined reliability CR values are between 0.828 and 0.872, both exceeding 0.7. Therefore, it indicates that the aggregated validity of the scale in this article is reliable. The specific inspection results are shown in Table 3.

The criterion for distinguishing validity is that the square root value of each latent variable AVE is greater than the correlation coefficient between that latent variable and other latent variables [4]. This article uses sample data to conduct discriminant validity tests and finds that the absolute value of the correlation coefficient between any two latent variables is less than the square root of the corresponding latent variable AVE, indicating a certain degree of discrimination between the latent variables in this article [4]. The specific analysis results are shown in Table 4. Therefore, the five latent variables of government support, resource

**Table 4. Results of discriminant validity test.**

| Variable | 1 | 2 | 3 | 4 | 5 |
|---|---|---|---|---|---|
| Government support | **0.749** | | | | |
| Resource endowment | 0.514 | **0.780** | | | |
| Industrial cluster | 0.553 | 0.572 | **0.713** | | |
| Regional culture | 0.389 | 0.491 | 0.509 | **0.741** | |
| Brand reputation | 0.525 | 0.538 | 0.579 | 0.590 | **0.794** |

Note: The diagonal is the square root of the AVE

empowerment, industrial clusters, regional culture, and brand reputation in this article are different constructs, and their discriminant validity is reliable.

## 4.4. Structural equation model fitting test

This article uses AMOS 24 software to construct a structural equation model of the formation mechanism of regional public brands for agricultural products, as shown in Fig 2. This article conducted a fitting test on the model and found that the ratio of chi square degrees of freedom is 1.868 (<3),GFI = 0.925 (>0.9), AGFI = 0.906(>0.9), IFI = 0.962(>0.9), CFI = 0.961(>0.9), TLI = 0.955(>0.9), RMSEA = 0.046 (<0.08) [11]. Therefore, the mechanism model for the formation of regional public brands for agricultural products constructed in this article has a good fit with the collected sample data.

## 4.5. Hypothesis testing

**4.5.1. Direct effect testing.** This article uses AMOS 24.0 software to conduct structural equation modeling analysis on the formation mechanism of regional public brands for agricultural products. When the CR value is greater than 1.96 and the p-value is less than 0.05, it is considered that the path coefficient has passed the significance test within the 95% confidence interval, indicating that the path is valid [29]. This article analyzes the direct effect pathways and finds that the critical ratio CR values of each pathway are all greater than 1.96, and the

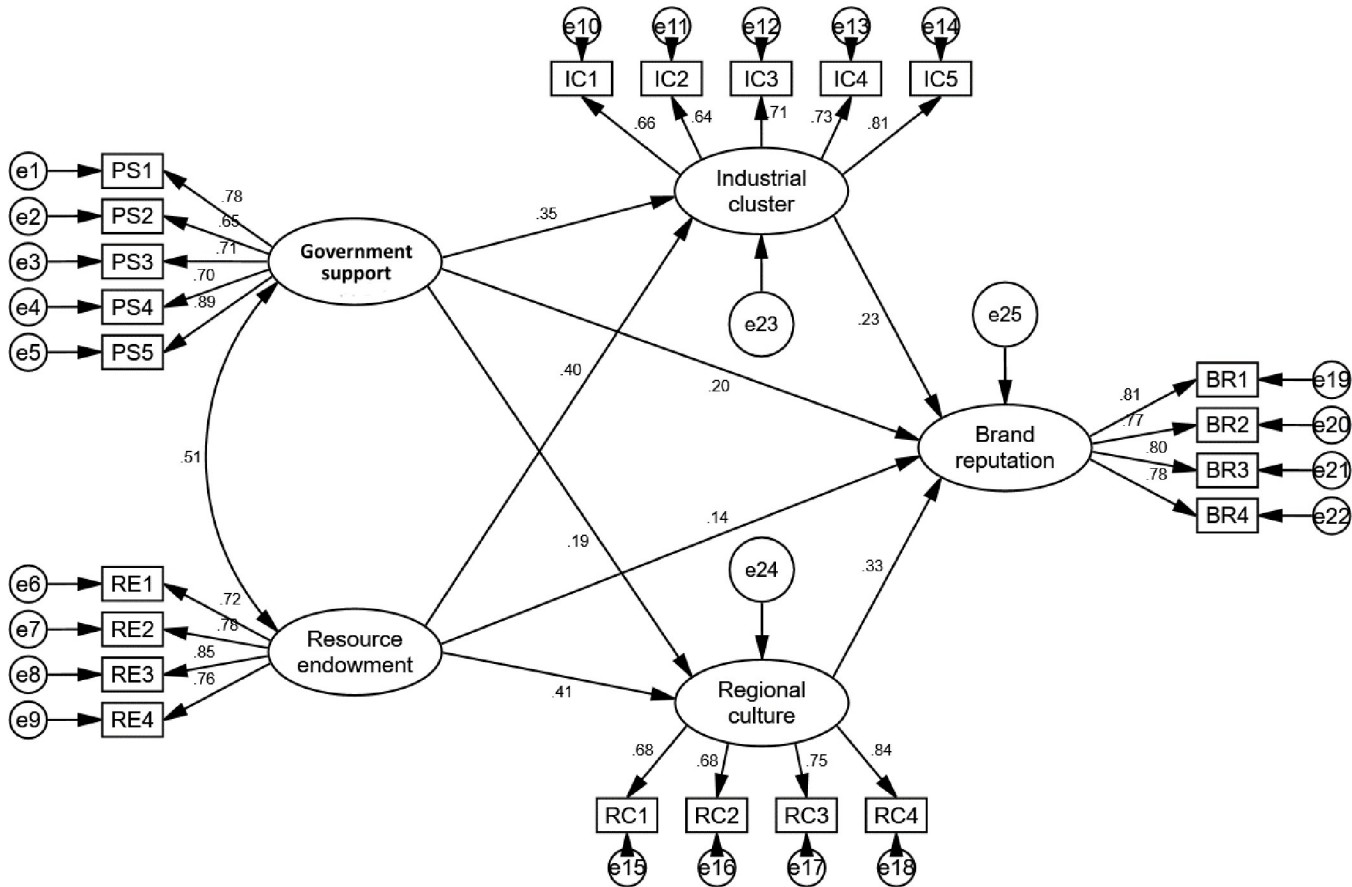

**Fig 2. Structural equation model of regional public brands for agricultural products.**

**Table 5. Direct effect test results.**

| No. | Path | Estimate | Std Estimate | S.E. | C.R. | P | Results |
|---|---|---|---|---|---|---|---|
| H1 | Government support→Industrial cluster | 0.301 | 0.349 | 0.053 | 5.687 | *** | establish |
| H2 | Resource endowment→Industrial cluster | 0.373 | 0.405 | 0.059 | 6.291 | *** | establish |
| H3 | Government support→Regional culture | 0.192 | 0.190 | 0.063 | 3.039 | 0.002 | establish |
| H4 | Resource endowment→Regional culture | 0.438 | 0.407 | 0.074 | 5.938 | *** | establish |
| H5 | Government support→Brand reputation | 0.208 | 0.196 | 0.063 | 3.311 | *** | establish |
| H6 | Resource endowment→Brand reputation | 0.163 | 0.144 | 0.075 | 2.182 | 0.029 | establish |
| H7 | Industrial cluster→Brand reputation | 0.284 | 0.231 | 0.080 | 3.532 | *** | establish |
| H8 | Regional culture→Brand reputation | 0.352 | 0.335 | 0.062 | 5.689 | *** | establish |

significance P values are all less than 0.05. The specific data of other values such as non-standardized coefficient (Estimate), standardized path coefficient (Std Estimate), and standard error (S.E.) are shown in Table 5.

According to the test results, the direct effect of this article is that assuming H1 is true, government support has a significant positive impact on industrial clusters($\beta$ = 0.349, p<0.001); H2 holds the resource empowerment has a significant positive impact on industrial clusters($\beta$ = 0.405, p<0.001); H3 holds that government support has a significant positive impact on regional culture ($\beta$ = 0.19, p<0.05); H4 confirms the positive impact of resource empowerment on regional culture is significant ($\beta$ = 0.407, p<0.001); H5 emphasis the government support has a significant positive impact on the reputation of regional public brands for agricultural products ($\beta$ = 0.196, p<0.001); H6 inspires that resource empowerment has a significant positive impact on the reputation of regional public brands for agricultural products ($\beta$ = 0.144, p<0.05); H7 holds that industrial clusters have a significant positive impact on the reputation of regional public brands for agricultural products($\beta$ = 0.231, p<0.001); H8 is established, indicating that regional culture has a significant positive impact on the reputation of regional public brands for agricultural products($\beta$ = 0.335, p<0.001).

**4.5.2. Mediation effect test.** This article uses the Bootstrapping method to conduct mediation effect testing. If the confidence interval does not include 0, it indicates a significant mediating effect; if it includes 0, it indicates that the effect is not significant [24]. This article sets a random sampling sample size of 2000 and a confidence interval of 95% [24]. Using the Bias Corrected estimation method, Amos calculated that the confidence intervals for each path were blocked above 0 and did not include 0. The specific results of other test values, such as the mediation effect value and standard error, are shown in Table 6.

According to the test results, the 95% upper and lower intervals of the mediating effect path of "government support→industrial clusters→brand reputation" in this article are [0.031, 0.149], excluding 0. This result indicates that industrial clusters have a significant mediating effect between government support and regional public brand reputation of agricultural products, with an effect value of 0.080. Therefore, hypothesis H9 is valid. The 95% upper and lower

**Table 6. Mediation effect test results.**

| No. | Path | Effect | S.E. | Bias-Corrected 95%CI | | Results |
|---|---|---|---|---|---|---|
| | | | | | | |
| H9 | Government support→Industrial cluster→Brand reputation | 0.080 | 0.030 | 0.031 | 0.149 | establish |
| H10 | Government support→Regional culture→Brand reputation | 0.064 | 0.026 | 0.020 | 0.123 | establish |
| H11 | Resource endowment→Industrial cluster→Brand reputation | 0.093 | 0.037 | 0.036 | 0.180 | establish |
| H12 | Resource endowment→Regional culture→Brand reputation | 0.136 | 0.033 | 0.082 | 0.212 | establish |

intervals of the "government support→regional culture→Brand reputation"mediation path are [0.020, 0.123], excluding 0. This result indicates that regional culture has a significant mediating effect between government support and regional public brand reputation of agricultural products, with an effect value of 0.064. Therefore, hypothesis H10 is valid. The 95% upper and lower intervals of the mediation path from "resource empowerment→industrial clusters →brand reputation" are [0.036,0.180], excluding 0. This result indicates that industrial clusters have a significant mediating effect between resource empowerment and regional public brand reputation of agricultural products, with an effect value of 0.093. Therefore, hypothesis H11 is valid. The 95% upper and lower intervals of the mediation path from "resource endowment→regional culture→brand reputation" are [0.082, 0.212], excluding 0. This result indicates that regional culture has a significant mediating effect between resource empowerment and regional public brand reputation of agricultural products, with an effect value of 0.136. Therefore, hypothesis H12 is valid.

## 5.Discussion

We empirically studies 416 samples of Yingde black tea and concludes that all research hypotheses are valid. Government support, resource empowerment, industrial clusters, and regional culture have a significant positive impact on the reputation of regional public brands for agricultural products, and industrial clusters and regional culture play a mediating role in the relationship between government support, resource empowerment, and the reputation of regional public brands for agricultural products. Based on empirical results, this article constructs a behavioral model for the formation mechanism of regional public brand of agricultural products, as shown in Fig 3. This model graph presents the direct effects between variables and the path coefficients obtained from regression analysis (β) [32,71].

H2, H4, and H6 indicate that resource empowerment has a positive impact on industrial clusters, regional culture, and regional public brand reputation for agricultural products. The formation of characteristic agricultural products is often based on regional natural resource empowerment, and gradually forms an industrial cluster with characteristic agricultural products as the core on the basis of resource empowerment [53,54,72]. The production and consumption process of characteristic agricultural products generated by resource empowerment will gradually form a regional characteristic culture related to it in the local area [72]. These will drive the development of regional public brands for agricultural products and contribute to the improvement of the reputation of regional public brands. Therefore, the research results on resource empowerment in this article are consistent with existing research findings.

H1, H3, and H5 indicate that government support has a positive impact on industrial clusters, regional culture, and the reputation of regional public brands for agricultural products. However, Yan and Yan jun found no significant correlation between government support and regional brand reputation of agricultural products [26]. The reason for this situation is that the research object of this article, Yingde Black Tea, is located in Guangdong Province, China. The local government has spent more financial expenses on cultivating regional public brands for agricultural products, and has played a very significant role in promoting the construction of regional public brands for agricultural products [73]. Guangdong Province is located at the forefront of China's economic reform and opening up of the country, and the government is also flexible and efficient in formulating policies related to regional public brands of agricultural products [73]. Yan and Yan jun's research object is the Xinjiang Turban grape. The Xinjiang government is facing relative financial difficulties and is unable to provide significant financial support to regional public brands of agricultural products [74]. Xinjiang is located in the inland hinterland of China, and the government is relatively conservative and not very

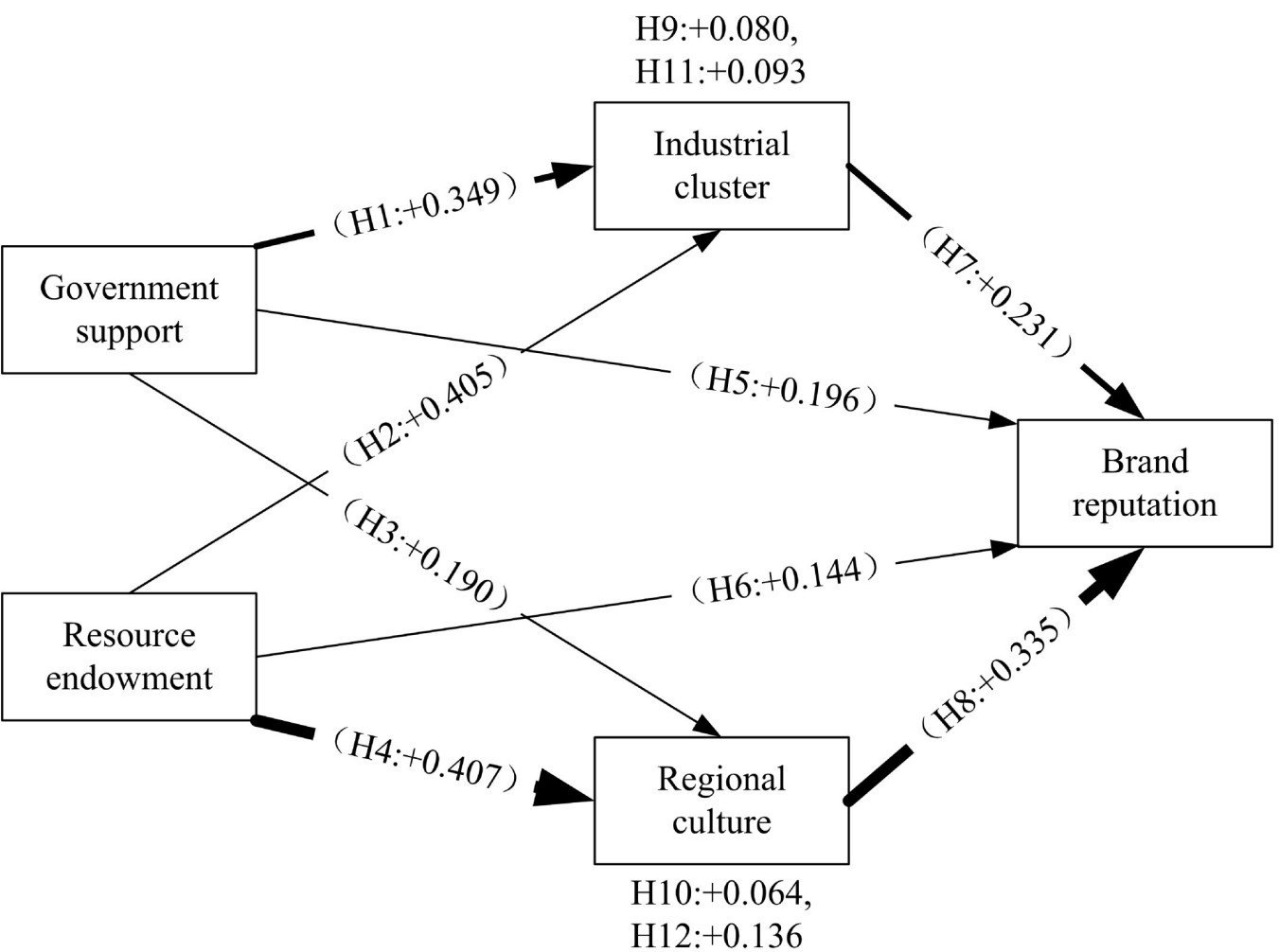

**Fig 3. Behavioural model of regional public brands for agricultural products.**

flexible in formulating relevant policies [74]. These have led to a lack of clear support from the Xinjiang government for regional public brands of agricultural products. Therefore, when the government provides sufficient financial support and flexible policies in the construction process of regional public brands for agricultural products, it can have a positive effect [28].

H9, H10, H11, and H12 indicate that industrial clusters and regional culture play a mediating role between government support, resource empowerment, and regional public brand reputation of agricultural products. Resource empowerment will influence the reputation of regional public brands for agricultural products through industrial clusters [52]. On this basis, this article expands and identifies the path through which government support and resource empowerment influence the reputation of regional public brands for agricultural products from a cultural perspective through regional culture. The research results of this article indicate that the construction process of regional public brands for agricultural products should not only focus on cultivating industrial clusters, but also on exploring and disseminating regional culture. The effect value of resource empowerment on the reputation of regional public brands of agricultural products through regional culture (0.136) is greater than the effect value of resource empowerment on the reputation of regional public brands of agricultural

products through industrial clusters (0.093). This is consistent with Keller's proposition that brand assets are built on brand knowledge in the minds of customers [9]. A regional culture based on unique resource empowerment helps regional brands form a unique brand personality image in the minds of consumers, thereby improving brand reputation [9]. The effect value of government support on regional public brand reputation of agricultural products through industrial cluster (0.080) is greater than the effect value of government support on regional public brand reputation of agricultural products through regional culture (0.064). This is because government support has a stronger impact on the construction of industrial clusters than on regional culture [44].

# 6.Summarize

## 6.1 Research conclusion

This article takes the regional public brand of Chinese Yingde black tea as the research object, and empirically studies the formation mechanism of a regional public brand for agricultural products by collecting 416 valid sample data. The research results indicate that government support, resource empowerment, industrial clusters, and regional culture have a significant positive impact on the reputation of regional public brands for agricultural products; government support has a positive impact on industrial clusters and regional culture; resource empowerment has a positive impact on industrial clusters and regional culture; industrial clusters and regional culture play a mediating role between government support, resource empowerment, and regional public brand reputation of agricultural products.

## 6.2 Theoretical contribution

The theoretical contributions of the research results in this article mainly include: (1) a comprehensive analysis of the interrelationships between government support, resource endowment, industrial clusters, regional culture, and regional public brands of agricultural products, revealing the formation mechanism of regional public brands of agricultural products. The existing literature mainly studies the single influencing factor or dual factor synergy of regional public brand reputation of agricultural products. The comprehensive analysis in this article has certain theoretical contributions. (2) Revealed the impact path of government support and resource endowment on regional public brands of agricultural products. This study found that government support and resource endowment not only directly affect the regional public brand of agricultural products, but also have an impact on the regional public brand of agricultural products through industrial clusters and regional culture. The mediating role of regional culture reveals the main pathways through which resource endowments play a role in the regional public branding of agricultural products. The intermediary role of industrial clusters reveals the main path through which government support plays a role in promoting regional public branding of agricultural products. (3) The research findings of this article are a beneficial supplement to the theory of regional branding. The existing regional brand theory is based on advertising and focuses more on the communication effects of brand formation process. This article studies the formation mechanism of regional public brands for agricultural products from the perspective of influencing factors, deepening the foundation of regional brand theory.

## 6.3 Practical inspiration

The research results of this article have certain enlightening effects on the practice of regional public brand construction for agricultural products. (1) Government support and resource

endowments have a significant impact on the construction of regional public brands for agricultural products. Therefore, when building regional public brands for agricultural products in various regions, on the one hand, local governments need to provide financial support, formulate preferential policies, and coordinate public resources to strongly support the construction of regional public brands for agricultural products; On the other hand, it is necessary to tap into local resource endowments, such as unique climate and soil, high-quality agricultural product varieties, advanced planting techniques, etc., to cultivate unique quality agricultural products, in order to form a unique advantage that regional public brands of agricultural products cannot be imitated in terms of product quality and cost. (2) Industrial clusters and regional culture play an intermediary role between government support, resource endowments, and regional public brands of agricultural products. The effect value of government support on the reputation of regional public brands of agricultural products through industrial clusters is greater than that of government support on the reputation of regional public brands of agricultural products through regional culture. Therefore, the government should focus on supporting the development of industrial clusters by cultivating industrial clusters with characteristic agricultural products as the core, promoting division of labor and cooperation within the agricultural product industry chain, improving the operational efficiency of agricultural product production and processing, and promoting technological innovation within the industrial clusters. The effect value of resource endowment on the reputation of regional public brands of agricultural products through regional culture is greater than that of resource endowment on the reputation of regional public brands of agricultural products through industrial clusters. Therefore, it is necessary to pay attention to the excavation and dissemination of regional culture based on resource endowment, forming a regional culture characterized by resource endowment, enriching the image of regional public brands for agricultural products, cultivating the personality characteristics of regional public brands, and thereby improving consumer cultural identity with regional public brands.

## 6.4 Research limitations and prospects

There are still certain limitations to the research in this article. Firstly, the sample is sourced from Yingde City, China, and the construction process of regional public brands for agricultural products involves government support. However, there are differences in the administrative methods of governments in different regions. Therefore, the research results are more in line with the construction laws of regional public brands for agricultural products in China. Secondly, we conducts research using cross-sectional data. Although the result has studied the interrelationships between different factors in the process of building regional public brands for agricultural products, it cannot reveal the causal relationships and dynamic mechanisms among different factors in the process of building regional public brands for agricultural products. Again, this study primarily explores the mechanisms of regional public brand development for agricultural products from a macro-level perspective, without incorporating micro-level individual psychological factors into the research. However, farmers' willingness to participate in the co-creation of regional public brands for agricultural products, as well as consumers' emotional interactions with these brands, both have an impact on the development of regional public brands for agricultural products.

In the future, research will be further strengthened in the following areas. Firstly, to improve the universality of the research results, a comparative analysis of the construction process of regional public brands for agricultural products in different regions will be conducted in the later stage, to further validate the research results. Secondly, long-term tracking studies will be conducted on Yingde black tea to reveal the causal relationships and dynamic

mechanisms involved. Again, we will also track and study the impact of regional public brands of agricultural products on the brand competitiveness of small and medium-sized enterprises in the region, to explore the effects of regional public brands of agricultural products on regional economic construction. Finally, the study will explore the specific impacts of individual psychological factors from a social psychology perspective on the process of developing regional public brands for agricultural products, aiming to further deepen the theoretical understanding of these brands.

## Supporting information

**S1 Appendix. Survey questionnaire.**
(DOCX)

**S2 Appendix. Data available.**
(XLSX)

## Author Contributions

**Conceptualization:** Jian Zhang, Xiaojun Ke, Songyu Jiang.

**Data curation:** Jian Zhang, Songyu Jiang.

**Formal analysis:** Jian Zhang, Songyu Jiang.

**Investigation:** Jian Zhang.

**Methodology:** Jian Zhang.

**Project administration:** Xiaojun Ke.

**Validation:** Jian Zhang, Xiaojun Ke.

**Visualization:** Jian Zhang.

**Writing – review & editing:** Xiaojun Ke.

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
