## [Decision Letter · Decision Letter 0]

26 Feb 2024

PONE-D-24-01490A Structural Equation Model to Access the Regional Public Brands of Agricultural Products: Case of Chinese Yingde Black TeaPLOS ONE

Dear Dr. Jiang,

Thank you for submitting your manuscript to PLOS ONE. After careful consideration, we feel that it has merit but does not fully meet PLOS ONE’s publication criteria as it currently stands. Therefore, we invite you to submit a revised version of the manuscript that addresses the points raised during the review process.

We look forward to receiving your revised manuscript.

Kind regards,

Innocent Senyo Kwasi Acquah

Academic Editor

PLOS ONE

When you resubmit, please ensure that you provide the correct grant numbers for the awards you received for your study in the ‘Funding Information’ section."

3. In the online submission form you indicate that your data is not available for proprietary reasons and have provided a contact point for accessing this data. Please note that your current contact point is a co-author on this manuscript. According to our Data Policy, the contact point must not be an author on the manuscript and must be an institutional contact, ideally not an individual. Please revise your data statement to a non-author institutional point of contact, such as a data access or ethics committee, and send this to us via return email. Please also include contact information for the third party organization, and please include the full citation of where the data can be found.

Reviewers' comments:

Reviewer's Responses to Questions

**Comments to the Author**

1. Is the manuscript technically sound, and do the data support the conclusions?

Reviewer #1: Yes

Reviewer #2: Partly

Reviewer #3: Yes

2. Has the statistical analysis been performed appropriately and rigorously? 

Reviewer #1: Yes

Reviewer #2: Yes

Reviewer #3: Yes

3. Have the authors made all data underlying the findings in their manuscript fully available?

Reviewer #1: No

Reviewer #2: Yes

Reviewer #3: No

4. Is the manuscript presented in an intelligible fashion and written in standard English?

Reviewer #1: No

Reviewer #2: Yes

Reviewer #3: Yes

5. Review Comments to the Author

Reviewer #1: This is very interesting research that uses the structural equation technique to explore regional brands of agricultural products taking as a case study the Yingde Black tea. The hypotheses and the theoretical framework are well justified. However, I have some minor comments that may be considered by the authors:

1. The model in Figure 1 is not an empirical model. It is a theoretical framework. That is, is a theory of potential drivers that represents the phenomenon under study. The empirical model is the statistical version of the theoretical framework. In this work, the empirical model is presented in Figure 2.

2. I am very surprised to find that the authors did not include a table with the items that form part of the constructs. It is difficult to evaluate the meaning of the constructs without this information. Therefore, I require the authors to include the items considered in the questionnaire. An example of how a table of this nature can be included is presented in Tables 1 y 2 in May et al. (2021). This reference is listed below.

3. The constructs of the theoretical model are well justified from the literature. However, many works that use structural equation modelling in other contexts have also consider socio-psychological drivers. I think this is relevant for this research because farmers’ willingness to produce Yingde Black tea may also be influenced by these considerations (influence from the opinion of people in the social network such as friend, colleagues, family members, etc.; need for status, etc). I suggest the following examples to explain the relevance of socio-psychological considerations in decision making, and why they have been ignored in this draft:

Deng, J. Sun, P., Zhao, F., Hana, X., Yang, G. and Feng, Y. (2016). Analysis of the ecological conservation behavior of farmers in payment for ecosystem service programs in eco-environmentally fragile areas using social psychology models. Science of The Total Environment 550: 382-390.

May, D., Arancibia, S. and Manning, L. (2021). Understanding UK farmers’ Brexit voting decision: A behavioural approach. Journal of Rural Studies 81: 281-293.

May, D.E., Arancibia, S., Wang, C., Hill, N., and Behrendt, K. (2023). Understanding young Chinese consumers’ preferences for foreign clothing brands: a behavioural approach. Asia Pacific Journal of Marketing and Logistics 35(12): 3032-3051.

4. The discussion can be enriched/improved by explaining the significant items that form part of the constructs. This has the potential to explore additional strategies to affect the regional public brand of agricultural products. An example of how the “empirical” model can be presented is shown in Figure 3 in May et al. (2021), listed above.

5. Two very minor points. References in the main body should not include the name initials. For example, the authors wrote (Kavaratzis M., 2006), and should be (Kavaratzis, 2006). On the other hand, the authors list some shortcomings at the end of Section 2.1. The ideas listed in this case should be separated by ; (semicolons) (see Page 3). Please check the grammar in the full document.

Reviewer #2: Overall, the article provides a comprehensive analysis of the factors influencing the formation of regional public brands for agricultural products, with a specific focus on Chinese Yingde black tea. However, there are several areas where revisions and improvements could enhance the clarity, coherence, and scholarly rigor of the manuscript:

Introduction: The introduction provides a clear overview of the significance of regional public brands for agricultural products and the specific focus on Chinese Yingde black tea. However, it could benefit from more precise delineation of the research objectives and hypotheses. Add more recent and relevant studies.

Literature Review: While the literature review briefly mentions the four main perspectives on the formation mechanism of regional public brands, it lacks depth and critical analysis of existing research. Consider expanding this section to include a more thorough synthesis of relevant literature, highlighting gaps and inconsistencies that the current study aims to address.

Methodology: The methodology section should provide more details regarding the survey questionnaire design, sampling strategy, and data collection process to ensure transparency and replicability of the study.

Clarify the rationale behind the choice of structural equation modeling (SEM) as the analytical technique and provide justification for the model specifications.

Results: Present the findings in a clear and organized manner, providing descriptive statistics and parameter estimates for each variable in the SEM model. Discuss any unexpected or counterintuitive findings and offer possible explanations or hypotheses for further exploration.

Discussion: The discussion section should provide a comprehensive interpretation of the empirical results in relation to the research objectives and theoretical framework. Compare and contrast the findings with existing literature, identifying areas of convergence or divergence and discussing their implications for theory and practice.

Highlight the contributions of the study and suggest avenues for future research to address remaining questions or limitations.

Conclusion: The conclusion should succinctly summarize the key findings, reiterate the study's contributions, and offer practical implications for stakeholders in real-world. Avoid introducing new information or ideas in the conclusion section, and focus on synthesizing the main insights derived from the study.

Minor Comments: Ensure consistency in terminology and citation style throughout the manuscript.

Proofread the manuscript for grammatical errors, typographical mistakes, and clarity of expression.

Reviewer #3: The paper studies how government support, resource endowment, industrial clusters, and regional culture affect the regional public brand of agricultural products based on regional brand theory, using Chinese Yingde black tea as an example. The topic is of great interest and reality, however, the academic contribution should be refined and highlighted. Here is some advice.

1. Authors' surnames and given names were misused for some papers, such as Juan & Jin 2022a, Yueli & Qiner, 2022, Yuhan et al., 2023, etc. Please double check the references.

2. In Introduction, a clear description of research gap could improve the article's strength as well as offer novelty. Are there really no comprehensive research about different factors of regional public brands? Why to choose these four factors?

3. To evaluate the effectiveness of regional public brand construction for agriculture products, the degree of popularity or acceptence should also be considered besides brand repution. Otherwise, it is not successful for agricultural products since there may be a condition that people who know the brand think it's great, but only a few people know the brand. Evenmore, people think it's a good brand, but they don't want to buy.

4. The hypotheses amony policy support, industrial cluster, resource endowment, and regional culture, as shown in Figure 1, should be well reconsidered. Will regional culture have an impact on policy support? Will the development of industrial cluster have an impact on related policy support?

5. All questions of the questionnaire are from literature? No new questions based on the proposed research were put forward?

6. Section 5 is mainly about the analysis and further discussion about statistical results, the theoretical and practical implications are not clearly stated. How can the proposed model and results drive theoretical research progress or guide industrial improvement? For instance, how to cultivate industrial cluster and regional culture? Which direction to guide?

7. According to the practical implication of Section 5.2, the construction of a regional public brand does not only need policy support from the local government, but also financial support, then the construction process is local government-dominated or industry/enterprises-dominated?

6. PLOS authors have the option to publish the peer review history of their article (what does this mean?). If published, this will include your full peer review and any attached files.

Reviewer #1: No

Reviewer #2: No

Reviewer #3: No

---

## [Author Response · Author response to Decision Letter 0]

4 May 2024

Response to Reviewers

Dear experts:

Thank you very much for your revision suggestions. These revision suggestions can be very effective in helping me improve the quality of my paper. I have carefully revised the paper based on your revision suggestions. The following are specific modifications.

Expert serial number Suggestion Modification status

Reviewer #1 1. The model in Figure 1 is not an empirical model. It is a theoretical framework. That is, is a theory of potential drivers that represents the phenomenon under study. The empirical model is the statistical version of the theoretical framework. In this work, the empirical model is presented in Figure 2.

 Change the "empirical model" in Figure 1 to "theoretical framework".

 2. I am very surprised to find that the authors did not include a table with the items that form part of the constructs. It is difficult to evaluate the meaning of the constructs without this information. Therefore, I require the authors to include the items considered in the questionnaire. An example of how a table of this nature can be included is presented in Tables 1 y 2 in May et al. (2021). This reference is listed below. Thank you very much for recommending the reference paper to me.

I originally considered that the project table had a lot of content and was difficult to layout, so I submitted the measurement table as an attachment to the journal.But as you said, the problem this brings is that readers find it difficult to evaluate the significance of the scale structure.Therefore, I modified Table 1 to present the specific content of the scale structure.A survey questionnaire is also attached at the end of the paper.

 3. The constructs of the theoretical model are well justified from the literature. However, many works that use structural equation modelling in other contexts have also consider socio-psychological drivers. I think this is relevant for this research because farmers’ willingness to produce Yingde Black tea may also be influenced by these considerations (influence from the opinion of people in the social network such as friend, colleagues, family members, etc.; need for status, etc). I suggest the following examples to explain the relevance of socio-psychological considerations in decision making, and why they have been ignored in this draft:

Deng, J. Sun, P., Zhao, F., Hana, X., Yang, G. and Feng, Y. (2016). Analysis of the ecological conservation behavior of farmers in payment for ecosystem service programs in eco-environmentally fragile areas using social psychology models. Science of The Total Environment 550: 382-390.

May, D., Arancibia, S. and Manning, L. (2021). Understanding UK farmers’ Brexit voting decision: A behavioural approach. Journal of Rural Studies 81: 281-293.

May, D.E., Arancibia, S., Wang, C., Hill, N., and Behrendt, K. (2023). Understanding young Chinese consumers’ preferences for foreign clothing brands: a behavioural approach. Asia Pacific Journal of Marketing and Logistics 35(12): 3032-3051.

 Your suggestion is correct. Many times, the willingness of farmers to grow Yingde black tea may also be influenced by these factors (opinions from people on social networks such as friends, colleagues, family, etc.; demands for status, etc.). However, this article focuses on the construction of regional public brands for agricultural products, involving the organizational level and not delving into the psychological level of farmers.So for now, social and psychological factors will not be included in the variables of structural equations in this study.

Studying regional public brands of agricultural products from a socio psychological perspective would be a very good choice, and therefore will be an important topic for our further in-depth research.We will conduct new research based on the research findings of Deng, J. (2016).This is presented in the future outlook of this article.

The research results of Deng, J. (2016), May, D. (2021), and May, D.E. (2023) have provided great inspiration for this study, providing both theoretical guidance and methodological demonstration.Therefore, all three papers have been included in the reference list in this article.

 4. The discussion can be enriched/improved by explaining the significant items that form part of the constructs. This has the potential to explore additional strategies to affect the regional public brand of agricultural products. An example of how the “empirical” model can be presented is shown in Figure 3 in May et al. (2021), listed above. Referring to Figure 3 of May et al. (2021), an "empirical" model was created in the discussion section of the paper.

 5. Two very minor points. References in the main body should not include the name initials. For example, the authors wrote (Kavaratzis M., 2006), and should be (Kavaratzis, 2006). On the other hand, the authors list some shortcomings at the end of Section 2.1. The ideas listed in this case should be separated by ; (semicolons) (see Page 3). Please check the grammar in the full document. Thank you very much for your careful review. There are indeed issues with these areas.Therefore, we have made the following modifications.

(1) The citation format has been modified according to the requirements of the journal.

(2) Use the separator ";" to separate the shortcomings of the parallel relationship.

(3) A consistency check was conducted on the professional terminology and citation style throughout the entire text, and modifications were made according to the requirements of the journal.

Reviewer #2 Introduction: The introduction provides a clear overview of the significance of regional public brands for agricultural products and the specific focus on Chinese Yingde black tea. However, it could benefit from more precise delineation of the research objectives and hypotheses. Add more recent and relevant studies. The following content has been added in the third paragraph of the introduction to clarify the research objectives of this article.

However, there has been a lack of research on the formation mechanism of regional public brands for agricultural products, as well as a lack of research on regional public brands for Chinese agricultural products. Therefore, this article will take Chinese Yingde black tea as the research object, explore the formation mechanism of regional public brands for agricultural products, study the influencing factors and interrelationships of regional public brands for agricultural products, and provide suggestions for building regional public brands for agricultural products based on this.

 Literature Review: While the literature review briefly mentions the four main perspectives on the formation mechanism of regional public brands, it lacks depth and critical analysis of existing research. Consider expanding this section to include a more thorough synthesis of relevant literature, highlighting gaps and inconsistencies that the current study aims to address.

 The literature review section has added recent relevant research literature from the four influencing factors of regional public brand construction, making the relevant literature more comprehensive and analyzing the controversies among different scholars, emphasizing the urgent issues and inconsistent viewpoints that need to be addressed in current research.

 Methodology: The methodology section should provide more details regarding the survey questionnaire design, sampling strategy, and data collection process to ensure transparency and replicability of the study. The survey questionnaire was submitted to the journal in the form of an appendix, which may have affected reading. This issue is indeed not conducive to readers understanding the entire text. Therefore, Table 1 has been modified to fully present the specific content of the scale structure.At the end of the paper, a survey questionnaire was also attached, and the original data was uploaded to the public data platform: Mendeley Data. The data's doi is 10.17632/9sys2yrwnp.1. URL of data: https://data.mendeley.com/drafts/9sys2yrwnp . These data can be obtained for free.

 Clarify the rationale behind the choice of structural equation modeling (SEM) as the analytical technique and provide justification for the model specifications.

 Add the following content to the "3.3. Data analysis" section to illustrate the basis for selecting a structural equation model.

Structural equation modeling is a multivariate statistical method based on variable covariance matrix for factor analysis and path analysis. It is mainly used to explore the potential relationships between multiple variables and comprehensively analyze the interaction mechanisms between variables (Tu Qihan, 2024). This article studies the formation mechanism of regional public brands for agricultural products, exploring the interaction between government support, resource endowment, industrial clusters, regional culture, and regional public brands for agricultural products.

Explain the normative requirements of the model in the text through the following content.

This article also conducted confirmatory factor analysis using AMOS 24 software. The smaller the ratio of chi square degrees of freedom(χ²/df), the higher the model's adaptability. Usually, a value less than 3 indicates good adaptability (Wen-e et al., 2021). When the root mean square error (RMSEA) of approximation is less than 0.08, it indicates good adaptation (Wen-e et al., 2021). The closer other indicators such as Comparative Fit Index (CFI) and Relative Fit Index (TLI) are to 1, the better the fit of the model. A value greater than 0.9 indicates better fit (Wen-e et al., 2021).

 Conclusion: The conclusion should succinctly summarize the key findings, reiterate the study's contributions, and offer practical implications for stakeholders in real-world. Avoid introducing new information or ideas in the conclusion section, and focus on synthesizing the main insights derived from the study.

 Thank you very much for your suggestion. The conclusion and theoretical contribution sections have been revised to provide a clearer and more explicit expression of the theoretical contribution of this article, and are presented separately in the form of 6.2. This can make it easy for readers to understand the theoretical contribution of this article.

 Minor Comments: Ensure consistency in terminology and citation style throughout the manuscript.

Proofread the manuscript for grammatical errors, typographical mistakes, and clarity of expression. Thank you very much for your careful review. As the original manuscript of this article was written in Chinese, there were some confusion and even errors in the translation of professional terms into English, such as "scale" and "survey questionnaire", "government support" and "policy support". This revision has unified these professional terms. All scales have been changed to survey questionnaires, and all policy support has been changed to government support.

Reviewer #3 1. Authors' surnames and given names were misused for some papers, such as Juan & Jin 2022a, Yueli & Qiner, 2022, Yuhan et al., 2023, etc. Please double check the references. This article has made revisions to the references and formatting requirements of PLOS one, as well as the format of the latest paper published in this journal.

Mockel S (2024) The macroeconomic money-nature nexus: Are growing money supplies a relevant obstacle on the way to an ecologically sustainable global economy? PLOS Sustain Transform 3(1): e0000095. https://doi.org/10.1371/journal.pstr.0000095

 2. In Introduction, a clear description of research gap could improve the article's strength as well as offer novelty. Are there really no comprehensive research about different factors of regional public brands? Why to choose these four factors? This suggestion is very helpful for this article.

This revision adds recent relevant research literature in the literature review section from the four influencing factors of regional public brand construction, making the relevant literature more comprehensive. It also focuses on analyzing the controversies among different scholars, emphasizing the urgent issues and inconsistencies in viewpoints that need to be addressed in current research. This reflects the research value and novelty of this article.

 3. To evaluate the effectiveness of regional public brand construction for agriculture products, the degree of popularity or acceptence should also be considered besides brand repution. Otherwise, it is not successful for agricultural products since there may be a condition that people who know the brand think it's great, but only a few people know the brand. Evenmore, people think it's a good brand, but they don't want to buy. Your viewpoint is correct. The effectiveness of brand building requires evaluating the level of customer recognition, identification, and acceptance.

The selection of brand reputation as the benchmark variable in this article is also based on this consideration. In the survey questionnaire, the measurement of brand reputation was set as brand awareness, brand image, brand evaluation, and customer repeat purchases.Perhaps the submission of the survey questionnaire in the form of an appendix to the journal has affected your reading. Therefore, I have modified Table 1 to present the complete content of the survey questionnaire, so that readers can better understand the measurement of brand reputation in this article. The following BR1-BR4 are measurement items for brand reputation.

The basis for selecting brand reputation as the school logo in this article is as follows.

Brand reputation is a concentrated reflection of consumer attitudes towards regional public brands, and is also considered the most important achievement in the construction of regional public brands (Juan et al., 2021). Moreover, the regional public brand of agricultural products, as a public asset within the region, mainly affects the sales of related enterprise products in the region through brand reputation (Yongdong, 2023).

BR1: Yingde black tea has a high brand awareness.

BR2: Yingde black tea has a unique brand image.

BR3: Yingde black tea has a good brand evaluation.

BR4: Yingde black tea has many loyal customers who repeat purchases.

 4. The hypotheses amony policy support, industrial cluster, resource endowment, and regional culture, as shown in Figure 1, should be well reconsidered. Will regional culture have an impact on policy support? Will the development of industrial cluster have an impact on related policy support?

 The questions you raised are precisely the focus of this article's research. When conducting literature review in this article, it was found that there is inconsistency in the views of existing research results on the relationship between regional culture, government support, and industrial clusters. Therefore, the idea of conducting this study emerged.

The research findings of this article verify the mediating role of industrial clusters between government support, resource endowment, and the reputation of regional public brands for agricultural products. Regional culture plays a mediating role between government support, resource endowment, and the reputation of regional public brands for agricultural products.

However, whether regional culture and industrial clusters have a reverse effect on government support was not taken into account in the initial research of this article. This will be an issue that needs further in-depth attention when conducting further research in this article. Thank you very much for raising this issue.

Based on your suggestion, I have made some modifications to the research hypothesis section to strengthen the research foundation related to the mediation hypothesis, and added the following content.

The development of industrial clusters will further strengthen government support for this industry and introduce more favorable policies (Zheng Ailin, 2024). The government helps promote the construction of regional culture by organizing exhibitions, celebrations, and food culture festivals for characteristic agricultural products (Yue Ju, 2024). The prosperity of regional culture will also lead the government to increase investment and promotion in the c

---

## [Decision Letter · Decision Letter 1]

23 Aug 2024

PONE-D-24-01490R1A Structural Equation Model to Access the Regional Public Brands of Agricultural Products: Case of Chinese Yingde Black TeaPLOS ONE

Dear Dr. Ke,

Thank you for submitting your manuscript to PLOS ONE. After careful consideration, we feel that it has merit but does not fully meet PLOS ONE’s publication criteria as it currently stands. Therefore, we invite you to submit a revised version of the manuscript that addresses the points raised during the review process.

We look forward to receiving your revised manuscript.

Kind regards,

Ricardo Limongi

Academic Editor

PLOS ONE

Journal Requirements:

Additional Editor Comments:

In the next version, please improve according to the request reviewer's request to "add a small section explaining the role of socio-psychological drivers in this topic, and also explain that one of the limitations of the study is that these drivers were not included. Please add an argument for this omission."

Reviewers' comments:

Reviewer's Responses to Questions

**Comments to the Author**

1. If the authors have adequately addressed your comments raised in a previous round of review and you feel that this manuscript is now acceptable for publication, you may indicate that here to bypass the “Comments to the Author” section, enter your conflict of interest statement in the “Confidential to Editor” section, and submit your "Accept" recommendation.

Reviewer #1: (No Response)

2. Is the manuscript technically sound, and do the data support the conclusions?

Reviewer #1: Yes

3. Has the statistical analysis been performed appropriately and rigorously? 

Reviewer #1: Yes

4. Have the authors made all data underlying the findings in their manuscript fully available?

Reviewer #1: (No Response)

5. Is the manuscript presented in an intelligible fashion and written in standard English?

Reviewer #1: Yes

6. Review Comments to the Author

Reviewer #1: The authors have addressed most of my observations. However, point 3 is not satisfied. I am not asking to include socio-psychological drivers in this research. What I ask, is to add a small section explaining the role of socio-psychological drivers in this topic, and also explain that one of the limitations of the study is that these drivers were not included. Please add an argument for this omission.

7. PLOS authors have the option to publish the peer review history of their article (what does this mean?). If published, this will include your full peer review and any attached files.

Reviewer #1: No

---

## [Author Response · Author response to Decision Letter 1]

2 Sep 2024

Dear Reviewers,

We sincerely appreciate the valuable suggestions you have provided for improving our manuscript. These recommendations are of great importance to the refinement of our work. In response, we have carefully revised the manuscript according to all the suggestions. The specific changes made are detailed below.

Reviewer #1 

Recommendation: The authors have addressed most of my observations. However, point 3 is not satisfied. I am not asking to include socio-psychological drivers in this research. What I ask, is to add a small section explaining the role of socio-psychological drivers in this topic, and also explain that one of the limitations of the study is that these drivers were not included. Please add an argument for this omission.

Response to Reviewers: Your suggestion is excellent. Through our literature review, we have indeed found that some scholars have explored regional public brands of agricultural products from a social psychology perspective. Based on this, we have added the following content to the manuscript.

(1) In the literature review section, we have added a discussion on the influence of social psychological factors on regional public brands of agricultural products and explained the reasons for not including them in this study, as detailed below.

Some scholars have explored the relationship between individuals and regional public brands of agricultural products from a social psychology perspective. Multifaceted interactions have a significant positive impact on farmers' willingness to participate in the co-creation of regional public brands for agricultural products, with psychological contracts serving as a mediating factor between these interactions and farmers' co-creation intentions(Juan, 2021). There is a close relationship between regional public brands of agricultural products and consumers' emotional attitudes, with both exerting mutual influence on each other(Fuyin, 2024). The greater the psychological distance between consumers and agricultural products, the higher consumers rate those products from that region(Li Xia, 2022). Jian, D. et al. explored farmers' willingness to protect the agricultural ecosystem from a social psychology perspective, based on the Theory of Planned Behavior. They found that farmers' behavior is significantly influenced by their willingness to protect the ecosystem, which in turn is significantly affected by their attitudes, subjective norms, and perceived behavioral control(Deng J, 2016). The psychological factors of farmers pertain to the micro-level individual psychological changes(Cheng Hong, 2023); however, this study primarily explores the mechanisms of regional public brand development for agricultural products from a macro-level perspective. Therefore, this study does not incorporate social psychological factors into the research process.

(2) In the limitations section, we addressed the omission of social psychological factors, as detailed below.

Again, this study primarily explores the mechanisms of regional public brand development for agricultural products from a macro-level perspective, without incorporating micro-level individual psychological factors into the research. However, farmers' willingness to participate in the co-creation of regional public brands for agricultural products, as well as consumers' emotional interactions with these brands, both have an impact on the development of regional public brands for agricultural products.

(3) In the future research section, we outlined plans for further exploring regional public brands of agricultural products from a social psychology perspective, as detailed below.

Finally, the study will explore the specific impacts of individual psychological factors from a social psychology perspective on the process of developing regional public brands for agricultural products, aiming to further deepen the theoretical understanding of these brands.

#Journal Requirements

Recommendation: Please review your reference list to ensure that it is complete and correct. If you have cited papers that have been retracted, please include the rationale for doing so in the manuscript text, or remove these references and replace them with relevant current references. Any changes to the reference list should be mentioned in the rebuttal letter that accompanies your revised manuscript. If you need to cite a retracted article, indicate the article’s retracted status in the References list and also include a citation and full reference for the retraction notice.

Response to Reviewers: We have reviewed all the references cited in the manuscript and found no evidence of any retracted publications.

---

## [Editor Report · Decision Letter 2]

6 Sep 2024

A Structural Equation Model to Access the Regional Public Brands of Agricultural Products: Case of Chinese Yingde Black Tea

PONE-D-24-01490R2

Dear Dr. Ke,

We’re pleased to inform you that your manuscript has been judged scientifically suitable for publication and will be formally accepted for publication once it meets all outstanding technical requirements.

Kind regards,

Ricardo Limongi

Academic Editor

PLOS ONE

---

## [Editor Report · Acceptance letter]

19 Sep 2024

PONE-D-24-01490R2 

PLOS ONE

Dear Dr. Ke, 

I'm pleased to inform you that your manuscript has been deemed suitable for publication in PLOS ONE. Congratulations! Your manuscript is now being handed over to our production team.

Kind regards, 

on behalf of

Professor Ricardo Limongi 

Academic Editor

PLOS ONE